# Molecular and Epigenetic Control of Aldosterone Synthase, CYP11B2 and 11-Hydroxylase, CYP11B1

**DOI:** 10.3390/ijms24065782

**Published:** 2023-03-17

**Authors:** Yoshimichi Takeda, Masashi Demura, Mitsuhiro Kometani, Shigehiro Karashima, Takashi Yoneda, Yoshiyu Takeda

**Affiliations:** 1Endocrinology and Metabolism, Kanazawa University Hospital, Kanazawa 920-8641, Japan; aldo_takeda@yahoo.co.jp (Y.T.); mkome@med.kanazawa-u.ac.jp (M.K.); 2Department of Hygiene, Graduate School of Medical Science, Kanazawa University, Kanazawa 920-1192, Japan; m-demura@med.kanazawa-u.ac.jp; 3Institute of Liberal Arts and Science, Kanazawa University, Kanazawa 920-1192, Japan; skarashima@staff.kanazawa-u.ac.jp (S.K.); endocrin@med.kanazawa-u.ac.jp (T.Y.); 4Endocrine and Diabetes Center, Asanogawa General Hospital, Kanazawa 920-0811, Japan

**Keywords:** aldosterone, cortisol, methylation, adrenal gland, hormone-producing adenoma

## Abstract

Aldosterone and cortisol serve important roles in the pathogenesis of cardiovascular diseases and metabolic disorders. Epigenetics is a mechanism to control enzyme expression by genes without changing the gene sequence. Steroid hormone synthase gene expression is regulated by transcription factors specific to each gene, and methylation has been reported to be involved in steroid hormone production and disease. Angiotensin II or potassium regulates the aldosterone synthase gene, *CYP11B2*. The adrenocorticotropic hormone controls the 11b-hydroxylase, *CYP11B1*. DNA methylation negatively controls the *CYP11B2* and *CYP11B1* expression and dynamically changes the expression responsive to continuous stimulation of the promoter gene. Hypomethylation status of the *CYP11B2* promoter region is seen in aldosterone-producing adenomas. Methylation of recognition sites of transcription factors, including cyclic AMP responsive element binding protein 1 or nerve growth factor-induced clone B, diminish their DNA-binding activity. A methyl-CpG-binding protein 2 cooperates directly with the methylated CpG dinucleotides of *CYP11B2*. A low-salt diet, treatment with angiotensin II, and potassium increase the *CYP11B2* mRNA levels and induce DNA hypomethylation in the adrenal gland. A close association between a low DNA methylation ratio and an increased *CYP11B1* expression is seen in Cushing’s adenoma and aldosterone-producing adenoma with autonomous cortisol secretion. Epigenetic control of *CYP11B2* or *CYP11B1* plays an important role in autonomic aldosterone or cortisol synthesis.

## 1. Introduction

Steroid hormones play a pivotal role in regulating blood pressure, cardiac function, water and electrolyte balance, and stress response [1,2,3,4]. Mineralo- and glucocorticoids are synthesized through de novo steroidogenesis in the adrenal gland. Aldosterone synthesis occurs in numerous tissues including cardiovascular tissues [5], the brain [6], adipose tissues [7], and the peripheral nerves [8]. Extra adrenal production of cortisol is reported in the immune system, skin, and intestine [9,10].

The adrenal gland utilizes cholesterol and lipoproteins for the biosynthesis of pregnenolone and the following steroids in the mitochondria. Some of the steps in steroidogenesis occur in microsomes (the endoplasmic reticulum). The adrenal cortex is able to de novo biosynthesize cholesterol [11,12].

The adrenal cortex is composed of three functional zones. The *zona glomerulosa,* the outer zone of the gland, expresses aldosterone synthase, CYP11B2, which catalyzes the synthesis of aldosterone [13]. The renin-angiotensin system (RAS) and potassium regulate CYP11B2 expression. The *zona fasciculata* produces cortisol. CYP11B1 (11-hydroxylase) is highly expressed in the zona fasciculata and is regulated by ACTH [12]. The innermost layer, the *zona reticularis*, expresses *CYP17* and produces DHEA and is also a source of androstenedione (A4, delta4), which is the primary adrenal androgen in some species (Figure 1). The hypothalamus–pituitary–adrenal axis, via a negative or positive feedback system, controls cortisol and DHEA production [14]. The 11Beta-hydroxylase gene, *CYP11B1* expression, is regulated by ACTH and a cAMP-regulated signaling pathway involving the CREB protein family [15]. Although *CYP11B2* and *CYP11B1* are highly conserved, there are significant differences between the *CYP11B1* and *CYP11B2* 5′upstream region, which may explain the different control of the mechanism of transcription [16].

## 2. Epigenetic Control of Gene Expression

Epigenetic changes are inherited modifications that are not present in the DNA sequence. Gene expression is regulated at various levels, not only in response to DNA modification. Histone acetylation modifications regulate gene expression [17]. Gene silencing is induced by DNA hypermethylation [18]. Gene expression is also regulated by RNA modifications which mediate RNA metabolism [19].

## 3. DNA Methylation

DNA methylation at the 5′-cytosine of CpG dinucleotides is a major epigenetic modification in eukaryotic genomes and is required for mammalian development [20]. It is associated with the formation of heterochromatin and gene silencing.

Dysregulation of DNA methylation of RAS genes has been involved in the pathogenesis of hypertension and cardiovascular diseases [21]. DNA methylation is established during usual development and disease progression. However, the DNA methylation pattern is in part dynamic in response to environmental changes [22,23]. Cardiovascular disorders, diabetes mellitus, and dyslipidemia, as well as lifestyle changes, dynamically affect DNA methylation.

## 4. Histone Modifications

Histone modification is an epigenetic modification characterized by the addition of an acetyl group to histone proteins, specifically to the lysine residue within the N-terminal tail [24]. This histone modification is catalyzed by histone acetyl transferases (HATs) or histone deacetylases (HDACs), which are associated transcription factors (TFs) [25,26]. Huang et al. [27] reported that histone demethylase lysine-specific demethylase 1 (*LSD1*) deficient rodents showed increased aldosterone production.

## 5. Epigenetic Control of *CYP11B2*

Figure 2 shows the CpG sites of the *CYP11B2* promoter region. In human and rodent adrenal glands, *CYP11B2* expression is regulated by not only RAS but also by endothelin and atrial natriuretic hormones [28,29,30,31]. Although potassium stimulates aldosterone biosynthesis [32,33,34], its pathophysiological roles are unclear. Angiotensin II and potassium can activate a number of *cis*-acting elements in the promoter of this gene, including the cAMP response element (CRE), nerve growth factor-induced clone B (NGFI-B) response element (NBRE-1), activating transcription factor 1 (ATF1), or CRE-binding protein 1 (CREB1) binding to Ad1/CRE, increasing *CYP11B2* transcription [35,36] (Figure 3).

We have reported that angiotensin II dynamically changed DNA methylation patterns in the *CYP11B2* promoter [37]. DNA methylation patterns are unstable in CpG in several circumferences [38]. However, where and how do changes in DNA methylation take place in non-CpG promoter sites? We reported that stimulatory signals of potassium treatment led to DNA demethylation around transcription factor binding sites and a transcription start site, where the chromatin structure was relaxed [39] (Figure 4). DNA demethylation was observed during two days of potassium treatment, while the highest level of demethylation was evident by seven days. The *CYP11B2* promoter demethylation increased gene expression [39] (Figure 5A).

The CREB1/ATF and NR4A family members lead to the activation of transcription. In our study, DNA demethylation, CREB1 recruitment, and chromatin relaxation at Ad1 were detected within two days after potassium treatment (Figure 5B). In contrast, Ad5 lagged two days behind Ad1 in chromatin accessibility. CREB1/ATF family members start chromatin remodeling by DNA demethylation at Ad1. CREB1/ATF family members may help NR4A family members initiate chromatin remodeling. This combination leads to gained gene expression with DNA demethylation about the transcription start site (TSS) in this gene. Cooperative action collectively undertaken by the CREB1/ATF family and NR4A family plays a pump-priming function in the control chromatin remodeling and DNA methylation in the *CYP11B2* promoter [37].

After potassium withdrawal, DNA methylation, NR4A1 (NGFI-B) recruitment, and chromatin accessibility at Ad5 immediately returned to normal levels. In contrast, DNA hypomethylation, CREB1 recruitment, and chromatin relaxation at Ad1 continued for several days after the stop of the stimulation. CREB1/ATF family members are retained at Ad1, acting to hold the DNA hypomethylation and chromatin relaxation. A memory of the potassium stimulation in the *CYP11B2* promotor is functioning by the epigenetic mechanism [39].

DNA methyltransferases (DNMTs), DNMT3A and 3B, establish and maintain DNA methylation. DNMT1 serves DNA methylation patterns through sequential rounds of cell division [40]. In our study, DNA demethylation of the *CYP11B2* promoter was associated with decreased DNA methylation activities. The balance between DNA demethylation and methylation activities is a major factor in the DNA methylation pattern.

A low-sodium diet or treatment with Angiotensin II increases *CYP11B2* mRNA levels and aldosterone production in the cardiovascular tissues as well as in the adrenal gland [41]. We reported that an angiotensin II infusion in rats decreased the methylation ratio of *CYP11B2* and increased the gene expression in the adrenal gland [37]. Treatment with angiotensin II in the cultured adrenal cells showed the same results. A low-salt diet induces hypomethylation of rat *CYP11B2* and increases *CYP11B2* mRNA levels parallel with aldosterone synthesis. A high-salt diet or treatment with a type 1 angiotensin II receptor blocker increases the methylation ratio of this gene. Taken together, angiotensin II is a major contributing factor to *CYP11B2* methylation. The rat zona glomerulosa transcriptome is changed by dietary sodium intake, involving more than 280 differentially regulated genes [42]. Nishimoto et al. [42] suggest that a change in salt intake affects the transcriptome by neurological responses as well as by RAS activation.

Aldosterone plays an important role in the pathogenesis of cardiovascular and renal disease in experimental and clinical studies [43,44,45,46]. Treatment with angiotensin-converting enzyme (ACE) inhibitors or angiotensin II type 1 receptor blockers (ARBs) for a long time increases plasma aldosterone to above pretreated levels, which is called “aldosterone breakthrough” [47]. This phenomenon has important clinical consequences, especially in congestive heart failure [48]. Involvement of various in vivo factors such as ACTH, electrolytes, endothelins, and angiotensin II type 2 receptor actions [49] have been proposed to explain this breakthrough phenomenon; however, details concerning the underlying mechanism remain unknown. We have reported that although both the direct renin inhibitor and ARB caused aldosterone breakthroughs, plasma endothelin levels were not increased [50]. Treatment with ARB influences *CYP11B2* methylation. It would be interesting to know whether or not the treatment with an ACE inhibitor or ARB for a long time influences the methylation status of *CYP11B2* and leads to aldosterone breakthrough.

Hughes-Austin et al. [51] reported that serum high potassium levels are associated with an increased risk for all-cause mortality independent of renal function or other cardiovascular risk factors. Weir et al. [52] reported that patiomer, a nonabsorbed potassium binder, decreased circulating aldosterone as well as serum potassium levels in patients with chronic kidney disease (CKD) taking renin-angiotensin system (RAS) inhibitors. These data suggest that potassium regulates aldosterone synthesis independent of RAS. It is interesting to look at whether treatment with patiomer prevents cardiovascular events in CKD patients. Sakthiswary et al. [53] reported that urinary potassium excretion was increased in patients with aldosterone breakthrough. Potassium may be important for aldosterone synthesis during treatment with RAS inhibitors. The pathophysiologic importance of epigenetic modification of *CYP11B2* by potassium should be further studied.

## 6. Epigenetic Modification of *CYP11B2* in Aldosterone-Producing Adenoma

Primary aldosteronism (PA) is recognized as a common secondary hypertension and accounts for approximately 5–15% of the hypertension population [54]. The most common clinical subtypes of PA are aldosterone-producing adenoma (APA) and bilateral adrenocortical hyperplasia [55]. We and others previously reported a lower degree of methylation of *CYP11B2* in APAs than in adrenal tissues or non-functioning adrenal adenomas. A negative correlation between the *CYP11B2* methylation ratio and mRNA levels was identified [37,56,57]. Di Dalmazi et al. [58] evaluated DNA methylation levels of *CYP11A1, CYP11B1, CYP11B2, CYP17A1, CYP21A2, HSD3B, NR5A1,* and *STAR* in benign adrenocortical tumors. They found that the methylation rates of *CYP11B2* were decreased in APAs compared with non-functioning adenomas. Epigenetic control of *CYP11B2* expression may play an important role in aldosterone synthesis in APA. Yoshii et al. [59] reported that the methylation rate in several CpG sites was lower in APAs than in non-functioning adrenocortical adenomas. They found no significant relationship between methylation rates and mRNA levels. They also reported that KCNJ5 mutation in APAs did not affect the methylation status. Nishimoto et al. [60] reported an interesting case of a patient with PA. The patient’s adrenal subcapsular aldosterone-producing cell clusters (APCCs) developed into nodules, which caused hyperaldosteronism. Some of the APCCs possess somatic gene mutations known to increase aldosterone synthesis [61,62]. These findings suggest that APCCs may play a part in the pathogenesis of PA. However, Omata et al. [63] reported that APCCs are frequent in the adrenal glands of nonhypertensive Japanese individuals in which somatic mutations (most commonly in the calcium voltage-gated channel subunit alpha1 D (CACNA1D)) were detected. We found the *KCNJ5* mutation in aldosterone-producing microadenoma and APCCs, in which methylation rates of *CYP11B2* were decreased compared with adjacent adrenal tissues. However, Di Dalmazi et al. [58] reported that the methylation status of *CYP11B2* did not differ markedly between APCCs and adjacent adrenal tissues or non-functioning tumors. Sun et al. [64] reported specific subgroups of APCCs with markedly variant distribution patterns of metabolites. Further study is necessary to clarify the mechanism of overproduction of aldosterone in the APCC and APA, including epigenome and metabolome.

Mineralocorticoid receptors (MRs) are expressed in cardiovascular tissues and the kidneys. MR antagonists (MRA) (spironolactone, eplerenone, esaxerenone) have been used for the treatment of PA [65,66]. Several papers have reported cases of idiopathic hyperaldosteronism with spontaneous remission during MRA therapy [67]. We have reported a case of APA with remission after long-term spironolactone therapy [68]. We compared the remission rate between spironolactone and eplerenone therapy in essential hypertension and found no difference between the two groups (unpublished data). Ye et al. [69] reported that spironolactone inhibited basal- and angiotensin-II-stimulated aldosterone synthesis in human adrenal cells. However, eplerenone did not inhibit aldosterone synthesis in H295R cells. We have reported that eplerenone inhibited tissue RAS [49]. The effect of MRA on the methylation of *CYP11B2* in the cardiovascular tissues as well as in the adrenal gland should be clarified.

## 7. Extra-Adrenal Mineralo- and Glucocorticoid Synthesis

Aldosterone synthesis at extra-adrenal sites is regulated by the RAS [70]. The mRNA of the StAR gene, *CYP11A*, 3β-hydroxysteroid dehydrogenase, *CYP21, CYP11B1,* and *CYP11B2* are expressed in blood vessels and the heart [71,72]. We found that the *CYP11B2* mRNA levels were lower in renal arteries than in the adrenal gland and a hypermethylation status was seen in renal arteries [37].

Briones et al. [7] reported that the aldosterone synthase gene and protein were detected in 3T3-L1 and mature adipocytes, which produce aldosterone basally and in response to angiotensin II. In 3T3-L1 “adipocytes”, angiotensin II increased the *CYP11B2* expression. Treatment with ARB or inhibitors of calcineurin blunted the angiotensin II effects. FAD286 (an aldosterone synthase inhibitor) inhibited adipocyte differentiation.

We previously reported that the expressions of protein and mRNA of the mineralocorticoid receptor in the peripheral nerve were equal to those in the kidney. The nerve conduction velocity (NCV) in diabetic rats was significantly improved by treatment with a mineralocorticoid receptor antagonist [73]. Mohamed et al. [8] reported that aldosterone immunoreactivity, *CYP11B2* gene expression, and MR protein were abundant in peptidergic nociceptive neurons of the dorsal root ganglia. Furthermore, aldosterone and *CYP11B2* were significantly upregulated in peripheral sensory neurons under inflammatory conditions. They also showed that inhibition of aldosterone synthesis in peripheral sensory neurons attenuated nociceptive behavior after hind paw inflammation.

Aldosterone synthesis and the *CYP11B2* gene expression are upregulated in cardiac tissues during hypertrophic cardiomyopathy (HCM), which are recognized as major HCM phenotype modifiers [74]. Aldosterone directly affects cardiac hypertrophy and fibrosis. We previously reported that aldosterone locally produced in cardiovascular tissues exerts its effects via paracrine or intracrine mechanisms [75]. Garnier et al. [76] reported that transgenic mice overexpressing *CYP11B2* in the heart showed coronary endothelium-independent dysfunction without hypertrophy. Alesutan et al. [77] showed *CYP11B2* expression in the human coronary arteries as well as smooth muscle cells. *CYP11B2* mRNA levels were higher in the aortic tissues of klotho-hypomorphic (*kl/kl*) mice than in control mice. Spironolactone ameliorated aortic osteoinduction occurred in adrenalectomized (*kl/kl*) mice. We have reported that the treatment with spironolactone improved cardiac hypertrophy in adrenalectomized hypertensive rats [78]. Yoshimura et al. [79] reported increased *CYP11B2* expression in the hearts of patients with cardiac failure. We found a clear association between the CpG methylation and the *CYP11B2* gene expression in the cardiac tissues of HCM [57]. We predict that DNA methylation at CpGs 1 and 2 is a key determinant of the *CYP11B2* mRNA levels in the heart. Hypomethylation of the *CYP11B2* promoter causes an aberrant increase in *CYP11B2* gene expression, which induces cardiac hypertrophy or cardiomyopathy [57]. The molecular mechanisms regulating the demethylation of CpGs 1 and 2 in the heart should be established.

Cortisol, a life-sustaining adrenal hormone, is an endogenous glucocorticoid (GC) that maintains human homeostasis. This hormone is synthesized from cholesterol in the adrenal cortex by five enzymatic steps, and CYP11B1 catalyzes the final step of cortisol biosynthesis [11]. Cortisol exerts its action through binding to a GC receptor (GR) expressed in a variety of organs, and regulates hydro-mineral metabolism, blood pressure, and carbohydrate, protein, and fat metabolisms [12]. Cortisol also serves a pivotal role in anti-inflammation and immunosuppression [80].

Extra-adrenal GC synthesis has been reported in blood vessels, the skin, the brain, the immune system, and the intestine [9,10,81,82]. Circulating GC levels often do not reflect local GC levels. An adrenalectomy eliminates serum GC but not in the hippocampus or cerebral cortex [83]. The potential clinical importance of tissue GC synthesis should be further clarified.

## 8. Epigenetic Control of *CYP11B1*

Excess cortisol causes various disorders. Cushing’s syndrome is caused either by excessive medication of cortisol-like compounds or by tumors, such as pituitary and adrenal adenomas, which express high levels of the cortisol synthase gene *CYP11B1* and thereby produce a high level of cortisol [84,85]. Previous reports have demonstrated the overexpression of *CYP11B1* in adrenal Cushing’s syndrome [86]. However, the molecular mechanism underlying the *CYP11B1* overexpression in adrenal Cushing’s syndrome remains unclear.

The DNA methylation inhibitor, 5′-aza-2 deoxycytidine, increases *CYP11B1* expression in the adrenocortical cells [87], which suggests that its expression is regulated by DNA methylation. Figure 6A shows the CpG sites of the *CYP11B1* promoter region. When we treated cultured adrenal cells with the cAMP analog, 2′-O-dibutyladenosine 3′, 5′-cyclic monophosphate (dibutyric cAMP; dbcAMP), *CYP11B1* mRNA levels were increased in parallel with a decreased DNA methylation ratio [88].

## 9. Epigenetic Modification of *CYP11B1* in Cortisol-Producing Adenoma

Cortisol-producing adenoma (CPA) expresses *CYP11B1* entirely but not *CYP11B2* [87]. Kubota-Nakayama et al. [85] reported that gene and protein expression of *CYP11B1* were increased in CPAs. We reported higher mRNA levels of *CYP11B1* in concomitant with a lower methylation ratio in CPAs compared to adrenal tissues or nonfunctioning adenomas [88] (Figure 6B). However, Di Dalmazi et al. [58] reported that the *CYP11B1* mRNA levels and methylation status did not differ between Cushing’s adenoma and non-functioning adrenal adenoma. According to previous studies, the heterogeneity of the molecular and gene abnormalities exist in Cushing’s syndrome or subclinical Cushing’s syndrome (SCS) [89], in which epigenetic regulatory mechanisms of *CYP11B1* play an important role in cortisol overproduction.

## 10. Epigenetic Modification of *CYP11B1* in Aldosterone-Producing Adenoma with Autonomous Cortisol Secretion

SCS is an adrenal incidentaloma with autonomous cortisol secretion. The current diagnostic criteria of SCS in Japan are proposed by Yanase et al. [90]. They reported 14.4% of patients with adrenal incidentalomas [91]. SCS is much more complicated with obesity, diabetes mellitus, hypertension, and cardiovascular diseases compared with non-functioning adrenal adenomas [92]. Katabami et al. [93] reported 26% of patients with PA had mild autonomous cortisol secretion in a recent Japanese large cohort study. They reported that PA with SCS increases renal complications compared to PA without SCS. Autonomous cortisol secretion in PA also contributes to metabolic risk or cardiovascular complications [94,95]. We found that six of the sixteen APAs evaluated were associated with autonomous cortisol secretion [88]. These APAs tended to be larger in size and associated with an increased prevalence in cerebrovascular diseases than APAs without autonomous cortisol secretion. The *KCNJ5* gene mutation was found in six APAs with autonomous cortisol secretion and eight of the ten APAs without autonomous cortisol secretion. The *CYP11B1* promoter region was less methylated in APAs with autonomous cortisol secretion than in those without autonomous cortisol secretion. These findings further suggest the significant role of DNA methylation of the *CYP11B1* promoter on gene expression.

Inoue et al. [96] recently reported the correlation between plasma aldosterone concentration and blood pressure in patients with SCS. We did not find any differences in the DNA methylation state of *CYP11B2* between APAs with autonomous cortisol secretion and those without autonomous cortisol secretion. The mechanism of aldosterone synthesis in SCS with hypertension should be clarified.

## 11. MicroRNAs (miRNAs) in Post-Transcriptional Regulation

There is increasing evidence that miRNAs play an important role in the regulation of *CYP11B1* and *CYP11B2* gene expression as well as for the derived proteins [97]. miRNAs are single-stranded noncoding RNA molecules of approximately 22 nucleotides. They target specific nucleotides on the mRNA of protein-coding genes and directly repress post-transcription [98,99]. Recently, the role of miRNAs was investigated with a focus on genes of the human CYP11B subfamily [12]. *Dicer1* is a key enzyme in miRNA maturation. It affects the function of miRNA miR-24, which binds to the 3′-untranslated region of *CYP11B1* and *CYP11B2* mRNAs [98]. Lenzini et al. [99] reported that components of the Wnt/-catenin pathway, which were downregulated by miR-23 and miR-43a, change aldosterone synthesis. Vetrivel et al. [100] reported that miR-1247-5p was upregulated in cortisol-producing adenoma (CPA). MiR-379-5p was upregulated in primary bilateral macronodular adrenocortical hyperplasia (PBMAH). Both miR-1247-5p and miR-379-5p targeted specific components in the WNT signaling pathway. Whether or not the silencing of *CYP11B2* or *CYP11B1* using siRNAs can be applied for treating PA or Cushing’s syndrome should be studied.

## 12. Epigenesis of the Other Steroid Hormone Synthase Genes

### 12.1. Steroidogenic Acute Regulatory Protein (StAR)

The epigenetic regulation of the *StAR* in the ovary is reported. Luteinizing hormone (LH) stimulation increases *StAR* gene expression and histone modifications are involved in its regulation. Methylation has been reported to be involved in the regulation of *StAR* gene expression by changes in the ovarian cycle [101].

### 12.2. Cytochrome P450 Family 11, Subfamily A, Polypeptide 1 (CYP11A1)

Okada et al. [102] examined methylation and histone modification of *CYP11A1* by acute stimulation of hCG in ovarian granulosa cells and reported that both were affected by hCG and thus involved in gene expression. In a rat model of multiple cystic ovary syndrome, hypomethylation of a portion of the CpG site of the *CYP11A1* promoter region has been reported [103].

### 12.3. Aromatase (CYP19A1)

An increased *CYP19A1* expression and hypomethylated state in the follicle are reported [104]. In the corpus luteum, *CYP19A1* is highly methylated and gene expression is low. CpG islands were found in the CRE (cAMP-responsive element) region, suggesting a relationship between cAMP-stimulated *CYP19A1* gene expression and methylation [105].

### 12.4. 17α-Hydroxylase (CYP17A1)

In humans, CYP17A1 plays an important role in cortisol biosynthesis, while in rodents, 3β-HSD is important for corticosterone biosynthesis. CpG islands are reported to be present in rodents but absent in humans, and methylation and gene expression are reported to be related in rodents. However, the homology of genes between humans and rodents is about 45% and they share a common regulatory mechanism [106]. It is possible that some kind of methylation regulatory mechanism exists in humans as well.

## 13. Conclusions

The gene expression of *CYP11B2* and *CYP11B1* in the adrenal gland is regulated by epigenetic modification. Salt intake and potassium influence the methylation of the *CYP11B2* gene. A negative correlation between DNA methylation and *CYP11B1* expression is seen in Cushing’s adenoma and APA with autonomous cortisol secretion. These results suggest that the epigenetic regulation of both *CYP11B2* and *CYP11B1* contributes to the pathogenesis of autonomous aldosterone and cortisol synthesis.

## Figures and Tables

**Figure 1 ijms-24-05782-f001:**
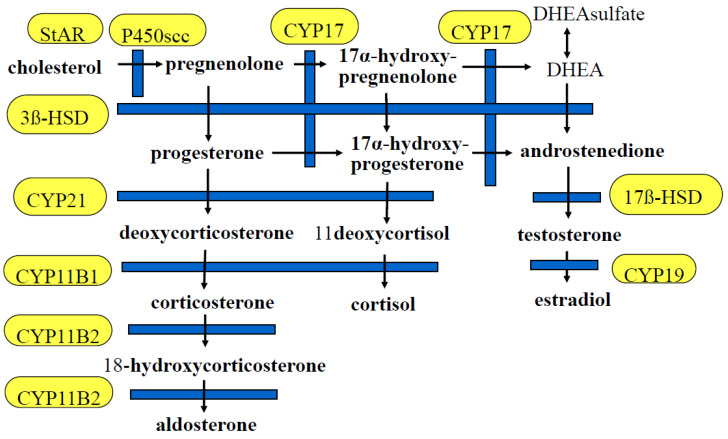
Steroid pathway. StAR, steroidogenic acute regulatory protein; P450scc, cholesterol side-chain cleavage enzyme; CYP17, 17α-hydroxylase; DHEA, dehydroepiandrosterone; 3β-HSD, 3β-hydroxysteroid dehydrogenase; CYP21, 21-hydroxylase; 17β-HSD, 17β-hydroxysteroid dehydrogenase; CYP11B1, 11 β-hydroxylase; CYP19, aromatase; *CYP11B2*, aldosterone synthase.

**Figure 2 ijms-24-05782-f002:**
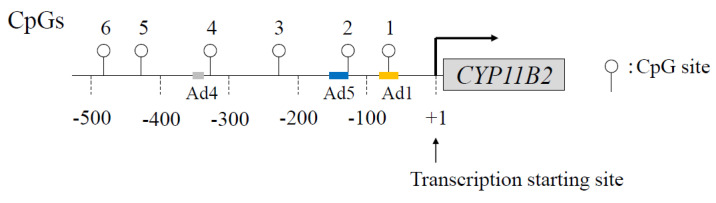
Schema of CpG dinucleotides within the human *CYP11B2* gene promoter. Open circle denotes CpG dinucleotides. Ad, cis-acting element.

**Figure 3 ijms-24-05782-f003:**
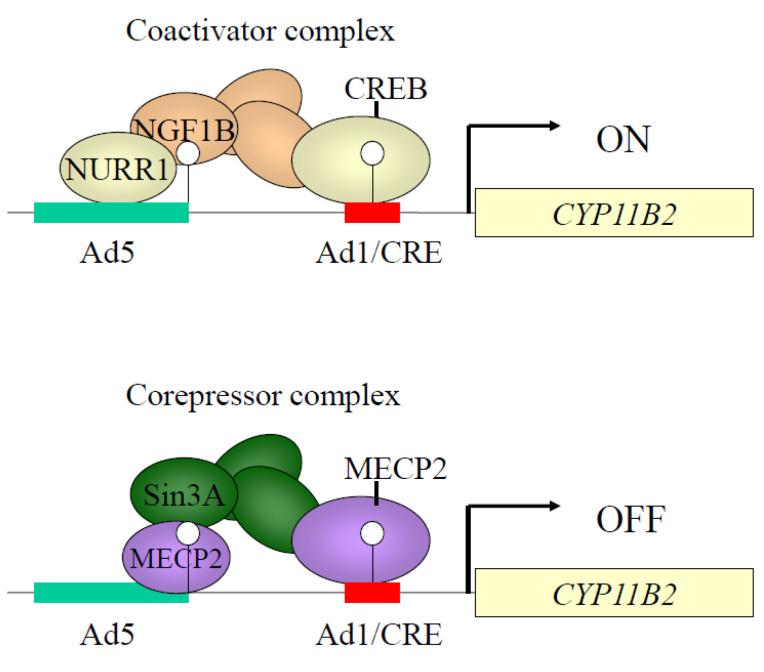
Coactivator and corepressor complexes of *CYP11B2* promoter region. Binding activities of coactivator complexes, such as CREB (cyclic AMP responsive element binding protein), NURR1 (nuclear receptor-related factor 1), and corepressor complex MECP2 (methyl-CpG-binding protein 2) are regulated by DNA methylation. Methylation of CpG1 greatly decreased CREB1 binding to Ad1 (cis-acting element 1). DNA methylation at CpG2 reduced basal binding activities of NGF1B (nerve growth factor-induced clone B) (NR4A1) and NURR1 (NR4A2) with Ad5. DNA methylation increased MECP2 binding to CpG1 and CpG2. NR4A, nuclear receptor 4 group A; Sin3A, SIN3 transcription regulator family member A.

**Figure 4 ijms-24-05782-f004:**
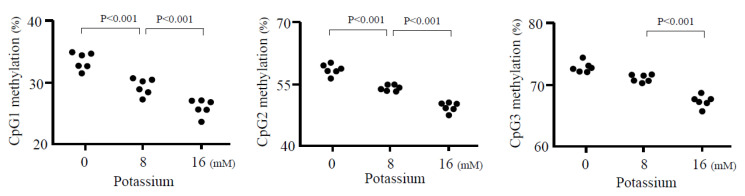
Effect of potassium on methylation of *CYP11B2* promoter in H295R cells. Potassium decreased in dose-dependent methylation ratio in CpG1, 2, and 3 sites.

**Figure 5 ijms-24-05782-f005:**
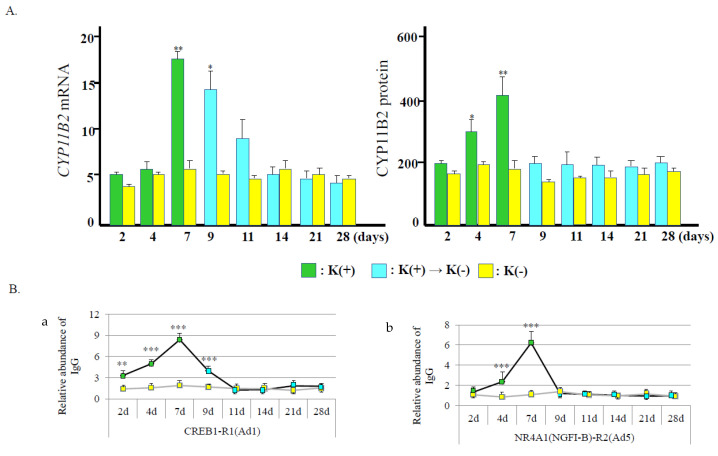
(**A**) Effect of potassium on *CYP11B2* mRNA and protein level. In *CYP11B2* mRNA level, results are given as fold change normalized to *ACTB*. * *p* < 0.01 and ** *p* < 0.005 vs. K(−). K(−) indicates H295R cells treated with no additional potassium. (**B**) Potassium-induced recruitment of CREB1 and NR4A1 in the *CYP11B2* promoter. (**a**), CREB1 recruitment to Ad1; (**b**), NR4A1 (NGFI-B) recruitment to Ad5. ** *p* < 0.005 and *** *p* < 0.0001 vs. K(−).

**Figure 6 ijms-24-05782-f006:**
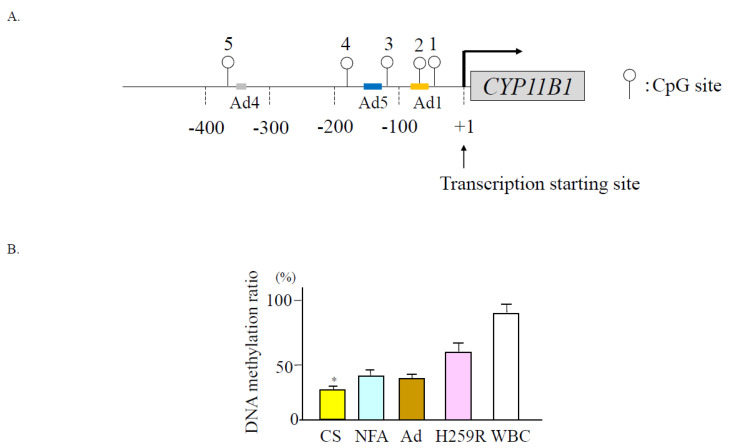
(**A**) Schema of CpG dinucleotides within the human *CYP11B1* gene promoter. Open circle denotes CpG dinucleotides (cited from Ref. [88]). (**B**) Methylation ratios of *CYP11B1* were significantly decreased in adenomas of Cushing’s syndrome. * *p* < 0.05 compared to other tissues; NFA, non-functioning adrenal adenoma; Ad, adjacent adrenal tissue; WBC, white blood cell.

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
