# Peer review of "Molecular and Epigenetic Control of Aldosterone Synthase, CYP11B2 and 11-Hydroxylase, CYP11B1"

_ijms, 2023, doi:10.3390/ijms24065782_

Round 1
Reviewer 1 Report
The manuscript by Takeda et al. entitled: "Molecular and epigenetic control of aldosterone synthetase, CYP11B2 and 11b-hydroxylase, CYP11B1" is a review that primarily summarizes the state of knowledge on the epigenetic control of gene expression involved in the synthesis of aldosterone (CYP11B2) and cortisol (CYP11B1) in the normal adrenal glands and aldosterone- and cortisol-producing adenomas.
At the beginning of the paper, steroidogenesis, DNA methylation and histone modification processes were discussed in general. In addition to epigenetic control of CYP11B1 and CYP11B2, the authors also briefly described the role of microRNAs in post-transcriptional gene regulation and epigenetic regulation of other selected genes of the steroidogenesis pathway. The whole concept of this work is interesting, the work is valuable and well written.
However, I have a number of minor comments regarding the form and content of the manuscript that should be considered before publishing the final version of this review.
Introduction
Lines 37-38: … Adrenal gland utilizes cholesterol and lipoprotein for the biosynthesis of pregnenolone and following steroids in the mitochondria…
In fact, some of the steps in steroidogenesis occur in microsomes (endoplasmic reticulum). Please complete the information.
Lines 44-45. … The innermost layer, the zona reticularis, expresses CYP17 and produces DHEA. (Fig. 1)…
The zona reticularis is also a source of androstenedione (A4, delta4), which is the primary adrenal androgen in some species. Please complete the information.
Figure 1. Androstendiol (proper name: androstenediol) is given incorrectly in the diagram. At this point in the steroidogenesis pathway, androstenedione (A4, delta4) is formed, not androstenediol. Please correct the diagram.
Line 55. Please add the word "enzyme" in the full name of P450scc.
Lines 54-57
Please put the letters alpha and beta in the names of the appropriate enzymes instead of the letters a and b, respectively.
Line 57. Replace "aldosterone synthese" with "aldosterone synthase".
Figures
It is not always clear where the figure comes from. Is it the authors' own scheme (Fig. 1, 2A, 3 and 5A) or does it come from a publication? Sometimes it is difficult to deduce from the text what is the source of a figure (e.g. Figure 4B). I suggest giving the source in the figure captions.
Figure 3 cannot be discussed before Figure 2B. It's against the rules of manuscript writing, and it's confusing. Please renumber the figures or change the text.
Other notes on the content of figures and figure captions
Figure 3. The description of the figure in the text (lines 85-87) does not fully match the graphic. On the other hand, sometimes the figure caption is not completely consistent with the text or the figure. In general, the relation of figure 3 to the text is unclear. Please describe Figure 3 in detail in the text (all proteins, regulatory factors, etc., please explain abbreviations).
Figure 2A. … Open circle denotes CpG dinucleotides… What do the white and red circles mean? Please explain in the description.
Figure 2A. What are Ad1-5? Please explain in the caption and in the text.
Figure 4A. Caption. I suggest K(-) instead of -K. The caption should match the chart.
In conclusion, all figures should be captioned so that there is no need to read the cited publication. If some information cannot be included in the caption, the full information should be included in the text of the manuscript.
Other remarks
Line 98. Replace A5 with Ad5.
Line 218. Replace the word hydorxysteroid with hydroxysteroid.
Line 224. Remove the dot after "adipocytes".
Line 281. Replace the word YP11B1 with CYP11B1.
Some abbreviations have not been explained in the text of the manuscript:
Line 48. CREB
Line 89. Ad1
Fig. 3. MECP2, Sin3A, NURR1
Line 123. TSS
Line 161 ACEI
Abbreviation list does not contain LH and hCG.
Author Response
Thank you very much for your review.
I changed my paper as your suggestion, Lines 37-38, Lines 44-45, Figure1, Line 55, Lines 54-57, Line 48, 89, 98, 218, 224, 234, and 281.
I explained following abbreviations, CREB, Ad1, MECP2, Sin3A, NURR1, TSS and ACEI.
I changed the number, Fig 2A to 2, Fig 2B to 4, Fig 4 to 5 and Fig 5 to 6.
I changed Fig 1, 2 and 3.
Fig 6A was cited from ref 88 and others are all original.
I changed the text of Fig 3 and all figures were captioned.
Reviewer 2 Report
Molecular and epigenetic control of aldosterone synthetase, CYP11B2 and 11β-hydroxylase, CYP11B1
Takeda et al. have presented a review that described the epigenetic mechanisms regulating expression of the steroidogenic enzymes aldosterone synthase (CYP11B2) and 11β-hydroxylase (CYP11B1). They have touched upon the various regulatory epigenetic machinery of CYP11B2 and CYP11B1 including DNA methylation, histone modifications and miRNAs. More importantly, they have described the various observations from multiple research groups with regard to the regulatory mechanisms for both these enzymes in primary aldosteronism (CYP11B2) and Cushing syndrome (CYP11B1). The authors have performed a vast survey of literature and presented a very concise review. A few minor corrections are needed.
1. Conventionally, the nomenclature for CYP11B2 is aldosterone ‘synthase’ and not ‘synthetase’.
2. Conventionally, CYP11B1 is not referred to as cortisol synthetase.
3. The authors need to be consistent with the use of ‘β’ vs. ‘Beta’.
Author Response
Thank you very much for your review.
I changed “aldosterone synthase”, 11b-hyddoxylase and b as your suggestion.